# Biomolecular Adsorprion at ZnS Nanomaterials: A Molecular Dynamics Simulation Study of the Adsorption Preferences, Effects of the Surface Curvature and Coating

**DOI:** 10.3390/nano13152239

**Published:** 2023-08-02

**Authors:** Roja Rahmani, Alexander P. Lyubartsev

**Affiliations:** Department of Materials and Environmental Chemistry, Stockholm University, S-10691 Stockholm, Sweden

**Keywords:** zinc sulfide, surface properties, biomolecular adsorption, molecular dynamics

## Abstract

The understanding of interactions between nanomaterials and biological molecules is of primary importance for biomedical applications of nanomaterials, as well as for the evaluation of their possible toxic effects. Here, we carried out extensive molecular dynamics simulations of the adsorption properties of about 30 small molecules representing biomolecular fragments at ZnS surfaces in aqueous media. We computed adsorption free energies and potentials of mean force of amino acid side chain analogs, lipids, and sugar fragments to ZnS (110) crystal surface and to a spherical ZnS nanoparticle. Furthermore, we investigated the effect of poly-methylmethacrylate (PMMA) coating on the adsorption preferences of biomolecules to ZnS. We found that only a few anionic molecules: aspartic and glutamic acids side chains, as well as the anionic form of cysteine show significant binding to pristine ZnS surface, while other molecules show weak or no binding. Spherical ZnS nanoparticles show stronger binding of these molecules due to binding at the edges between different surface facets. Coating of ZnS by PMMA changes binding preferences drastically: the molecules that adsorb to a pristine ZnS surface do not adsorb on PMMA-coated surfaces, while some others, particularly hydrophobic or aromatic amino-acids, show high binding affinity due to binding to the coating. We investigate further the hydration properties of the ZnS surface and relate them to the binding preferences of biomolecules.

## 1. Introduction

Interesting physical and chemical characteristics of nanomaterials are sometimes not visible in their bulk forms [1]. Moreover, modifications in nanoparticles’ size, shape, and coating affect their physical and chemical properties [2,3,4]. II-VI semiconductor nanomaterials, which are usually known as quantum dots (QDs), have drawn a lot of interest because of their unique optical properties including size-tunable photoluminescence in the visible spectral range, wide absorption, and sharp emission bands, which makes them useful for solar cells, electronic devices, optoelectronic, and biomedical purposes. Due to the quantum confinement effect, it is relatively easier to tune the frequency range of emitted light from semiconductor nanostructures. Compared to organic dyes [5] and fluorescent proteins, semiconductor QDs are more resistant to photobleaching, are brighter, can be excited with a single light source, and have multicolor emission depending on the core size [6]. These special features, producing different optoelectronic properties by controlling their sizes and shapes, together with their broad absorption, luminescent efficiency, high stability against photobleaching, and high chemical stability have made them an ideal candidate for biomedical purposes [7,8,9]. For example, in fluorescent labeling and imaging [10,11] (in vitro), which is one of the most common applications, QDs can be conjugated to a variety of targeting ligands, such as antibodies, peptides, small molecules, and nucleotides. Through a particular interaction between the target molecule and QDs, the location of the target molecule and its abundance can be revealed through fluorescence and molecules will be identified in this way. This could be useful for the delivery of anticancer drugs and fluorescent labeling of cellular proteins. Chemical- and photo-stability of QDs make them useful for in vivo imaging, cell lineage tracking, and gene therapy. They are excellent sensors [12] for sensing and detection because of their distinctive optical properties, such as brightness and a large Stokes shift, which can increase the detection sensitivity.

Although the special advantages of QDs for bioapplications are well recognized, there are still ongoing discussions about their potential toxicity in biological environments. Various studies have shown that nanoparticles exposed to the body through ingestion, inhalation, skin contact, etc., may cause serious health problems [13]. Generally, the capability of nanomaterials to induce the production of reactive species, cytotoxicity, genotoxicity, and neurotoxicity are the most common mechanisms, which cause adverse health effects [14]. For instance, cadmium-based semiconductors (CdS, CdSe, CdO, CdTe) have shown interesting optoelectronic properties due to their suitable band gaps and high absorption coefficients. They have wide applications in solar cells, light-emitting devices, photo-catalysts, biosensing of cancer cells, antibacterial and antifungal effects, etc. However several studies have demonstrated their highly cytotoxic, genotoxic, and carcinogenic effects in biological environments [15,16,17]. Besides the chemical composition, the physicochemical properties of nanomaterials such as the size of the nanoparticle, the surface area, shape, aspect ratio, surface coating, crystallinity, dissolution, agglomeration, etc., determine their biological effects including toxicity [18]. For instance, Liu et al. realized that by considering certain biological markers, the genotoxicity, and cytotoxicity of smaller CdS nanodots (3–5 nm) is higher than nanorods (30–50 nm) [19].

Surface coating of nanomaterials was suggested by researchers as a way to tailor their stability, wettability, dissolution, and functionality, as well as to minimize their effects on the biological environment. The coating may result in converting noxious particles to less toxic ones, but may also have the opposite effect, e.g., due to increased bioavailability. Sue et al. showed in cell viability studies that CdTe QDs have a cytotoxic effect leading to cellular disruption, but no cytotoxicity was observed in the coated form of CdTe/CdS/ZnS QDs [20]. Baranowska et al. observed that the ZnS layer coating ZnO inhibited the ZnO dissolving process in water and improved the mechanical and electrical properties of ZnO thin films [21].

QDs are often composed of semiconductor core (e.g., CdSe, CdTe), and are encapsulated by a shell (e.g., ZnS) with a wider band gap to enhance optical and electronic properties and reduce core metal leaching. This alone is not sufficient to stabilize the core, particularly in biological solutions, but the additional coverage of ZnS shell with polymers or ZnS shell silanization provides an increase in QD stability and a reduction in non-specific adsorption. Due to their enhanced structural, electrical, dielectric, optical [22], and thermal capabilities [23], organic-inorganic nanoscale composites have been identified as a distinct class of hybrid materials. There are several studies about nanocomposites made of ZnS nanoparticle polymers. For instance, Cheng et al. synthesized a zinc blende type of ZnS nanoparticles by coordination control with diethanolamine in ethylene glycol [24]. Beaune et al. explored the luminescence quenching of CdSeTe/ZnS nanoparticles coated with polyethylene glycol in the presence of Cu cations and compared them to InP/ZnS quantum dots coated with mercaptoundecanoic acid [25].

Choosing a proper molecule to coat the surface could be challenging because it might affect the application of nanomaterials. Since ZnS nanoparticles are optically active in the UV-visible range, surface-coating material for ZnS nanoparticles requires UV absorption below the ZnS absorption edge in order to enable photoluminescence. Due to this strict criterion, the majority of surfactants and polymers are not included on the list of acceptable capping agents for ZnS nanoparticles. One of the materials that meet the aforementioned criterion is poly methyl methacrylate (PMMA) [26,27]. Studies conducted by Ajibade et al. revealed the dispersion of ZnS, Cd, and HgS nanoparticles in PMMA matrices, and PMMA acted as a proper host matrix since the shape and properties of the dispersed semiconductor nanoparticles have not been affected [28]. PMMA has excellent transparency, good abrasion resistance, hardness, ultraviolet resistance, and stiffness. Due to its attractive optical and electrical properties [29,30], PMMA is widely used in many applications such as lenses, light pipes, skylights, light emitting devices, and biomedical application [31]. PMMA is a biomaterial that has been utilized in clinical orthopedic practice and dental implants [32,33] for many years because of its non-degradable, hard, strong, rigid, and transparent character. The PMMA can also serve as a spacer and as a delivery vehicle for antibiotics [34,35].

In this paper, we address the question of biomolecular adsorption on pristine and PMMA-coated ZnS surfaces, which is of high importance for both optimization of QDs functionality in biomedical applications, and for the understanding of molecular phenomena behind biological effects of nanoparticles and potential toxicity [36]. Biomolecular adsorption on nanoparticles typically follows typically physisorption mechanisms when biomolecules adsorb on the surface due to electrostatic (including hydrogen bonds) and van-der-Waals interaction. It was discussed previously that adsorption energies of biomolecules to a nanomaterial represent “biological fingerprints”, which are predictive of the biological activity of the nanomaterial and can be used, e.g., for prediction of protein corona composition and further fate of the nanomaterial in a living organism [37,38]. Experimentally it is very challenging to investigate molecular properties at the nanoparticle surfaces in the aqueous phase because of the very small volume of this area relative to the bulk. Computer simulations and particularly molecular dynamics (MD) can provide valuable information, and there are many examples of using molecular modeling to obtain insight into interface properties and adsorption on the atomistic level [39,40]. Specific challenges of MD simulation studies of biomolecular adsorption at inorganic surfaces and ways to overcome or alleviate these are discussed in a recent paper by Michaelis et al [41]. Previously, Nawrocki et al. [42] performed umbrella sampling MD simulation to study the interactions of 20 capped amino acids with ZnS (110) surface using an Amber99 force field. They found that the binding energy of adsorbed amino acids does not exceed 4.3 kJ/mol except for cysteine, for which special Morse potential between sulfur atoms of cysteine and ZnS slab was introduced. Hosseinzadeh et al. studied the interaction between insulin and ZnS QDs [43] functionalized with sulfur-contained molecules which have different functional groups (OH, COOH, NH${}_{2}$) and amino acids including cysteine (COOH and NH${}_{2}$). There exists also several experimental and theoretical studies for the interaction of biomolecules with other inorganic nanomaterials such as TiO${}_{2}$ [36,44,45,46], Au [47,48,49], ZnO [50,51,52], Si and GaAs [53,54].

The aim of this work is to provide a detailed description of biomolecular interaction with pristine and PMMA-coated ZnS nanosurfaces using enhanced-sampling molecular dynamics (Metadynamics). By considering both plain (slab) nanosurfaces and spherical nanoparticles, we investigate the curvature effect. The role of coating is investigated by simulations of both pristine and PMMA-coated surfaces. In our study, we use a modification (described below) of a force field developed by Namsani et al. [55], which was specifically parameterized to describe the interaction of ZnS with PMMA and eventual organic/biomolecules in aqueous media. To characterize biomolecular interactions with ZnS, we selected a set of 29 small biomolecular fragments representing side chain analogs of amino acids, fragments of lipids, and sugars. The same set of biomolecules was used in previous studies of TiO${}_{2}$ [36] and carbon-based [56] nanomaterials, and results of these studies were further used in training of machine learning algorithm predicting adsorption free energies for a much larger variety of organic molecules [57]. Furthermore, adsorption free energies of biomolecules to a specific material represent an important set of descriptors of bio-nano interface which is further used in nanoinformatics characterization of nanomaterials with respect to their functionality and safety [38].

## 2. Simulation Models and Methods

### 2.1. Models

To study the effects of shape and surface curvature of ZnS nanostructures on their interaction with biomolecules, we created a 2D periodic slab of ZnS crystal with (110) plane exposed to the solvent and a spherical ZnS nanoparticle. Both models were investigated in pristine form and coated with PMMA polymers. The systems were fully hydrated and in each simulation, the potential of mean force (PMF) and adsorption free energy of selected biomolecule to the ZnS surface was computed. The initial atomistic configurations of ZnS nanoparticles, ZnS (110) slabs, PMMA polymer, solvent, and biomolecules were obtained in the following way.

ZnS (110) Slabs

The most stable crystal form and surface cleavage of ZnS at room temperature is the zinc blende cubic form and (110) plane. The slab of ZnS (110), of size 4.33 × 4.59 × 3.32 nm, was constructed by replication of the ZnS unit cell. The slab is periodic in *X* and *Y* directions, it is uncharged and has 2688 atoms including 1344 Zn and 1344 S atoms. We have used this slab model for all slab-containing systems (pristine and coated forms) in this study.

ZnS Nanoparticles 

A spherical ZnS nanoparticle was made by taking periodically repeated cubic unit cells and cutting a sphere with a diameter of 5 nm. Single-coordinated atoms were then removed from the surface and the final radius of the nanoparticle become about 2.4 nm. The nanoparticle is uncharged and consists of 1800 Zn and 1800 S atoms. This model has been used in all nanoparticle-containing systems both in pristine and coated forms.

Biomolecules 

The 29 biomolecules we used as adsorbents in this study are the building blocks of proteins, lipids, and sugar. These include naturally occurring 18 amino acid side chain analogs except glycine and proline (the backbone fragment of amino acids replaced with hydrogen), glycine and proline including protein backbone with neutral terminals, ionic forms of histidine and cysteine, four fragments of lipid headgroup, and glucose, see the full list in Figure 1.

Coating 

We modeled PMMA polymer as consisting of three oligomers of methyl methacrylate. Each PMMA molecule consists of 53 atoms (Figure 2b). The number of PMMA molecules was chosen with the aim to cover the whole surface with about one full layer of PMMA. We started with an initial evaluation of the ZnS surface area and size of a PMMA molecule, added them to the system, and run a short simulation observing the adsorption of PMMA molecules to the surface. If the surface of the nanoparticle upon visual inspection still had some open space, we progressively added polymer to the system until it was evenly distributed across the surface of the nanoparticle. In the case of the slab system, PMMA was added to one of the ZnS surfaces, thus creating a coating layer at that surface, while the other surface remained uncoated. This was performed to speed up production computations for biomolecular interactions since interaction with just one surface is of interest. In total, 250 PMMA molecules were added to the spherical nanoparticle and 70 to the slab system.

Next, we hydrated the system by filling the rest of the simulation box with water. Furthermore, Na^+^ and Cl${}^{-}$ ions were added to provide a physiologically relevant ion concentration of 0.15 M. The composed in this way simulation systems, as well as their components, are shown in Figure 2.

### 2.2. Force Fields

For ZnS nanomaterials, we adopted the force field developed by Namsani et al. [55], which in the original formulation is a polarizable (within core-shell model) force field with Buckingham and harmonic 3-body interactions between Zn and S atoms of ZnS crystal, and Lennard–Jones potentials describing interactions of ZnS with the environment. We made some modifications to this force field making it suitable to simulate with the Gromacs simulation package. While in the Namsani’s force field, Zn and S atoms interact with each other by Buckingham potential and are non-bonded, we considered the Zn and S atoms as bonded in the structure of ZnS nanomaterials arguing that at the studied conditions, the ZnS nanostructures remain stable and can be well described by a bonded potentials providing correct structure. To obtain parameters of the bonded potential, we considered the total pair potential between Zn and S atoms in Namsani’s model, which is a sum of the Buckingham and Coulombic potentials, and we fitted it by a harmonic potential in the region of the expected Zn–S bond lengths between 2.0 and 2.5 Å. Furthermore, we substituted the 3-body potential of work [55] by the harmonic angular potential, which is fitted to the 3-body potential taken at a distance corresponding to the distance between nearest S atoms in the crystal structure of ZnS (3.4 Å). We have used the same charges (−2e for S and +2e for Zn) as in the force field [55] but we excluded polarizability of the core-shall model by considering S atoms as a single site with fixed charge. Also, electrostatic and Lennard–Jones interactions were excluded between bonded atoms. We have tested that this bonded potential model keeps well the original ZnS crystal structure with root mean square deviations (RMSD) from the crystal structure within 0.4 Å (Appendix A).

For PMMA–ZnS interactions, we substituted Buckingham potential describing the interaction of Zn and S atoms with PMMA oxygen double bonded to C atoms by the Lennard–Jones potential with appropriate parameters. All other interactions (PMMA except carboxyl oxygen, biomolecules ) were described by the GAFF force field [58], with force field types and partial charges assigned by *acpype* utility [59] using AM1-BCC method. For water, the TIP3P model was used. The cross interaction LJ parameters are calculated using the Lorentz–Berthelot combination rules: $\sigma_{i}=\left(\sigma_{i}+\sigma_{j}\right)/2$ and $\epsilon_{ij}={\left(\epsilon_{i}\epsilon_{j}\right)}^{1/2}$. A summary of the specific ZnS and PMMA force field parameters used in this work is given in Table 1.

### 2.3. Methods

#### 2.3.1. Free Energy Calculations

Adsorption of molecules at the surface is characterized by the adsorption free energy, which is a difference between the free energies of a molecule in the bound and free state
(1)$$\Delta G_{ads}=G_{bound}-G_{free}$$
where $G_{bound}$ and $G_{free}$ correspond to free energies associated with a molecule attached to a surface and a molecule positioned far away from the surface, where surface interactions are minimal. Adsorption free energy is related to the potential of mean force (PMF), which can be interpreted as the free energy of the solute at distance *s* from the surface, and which can be determined as $W\left(s\right)=-k_{B}T\ln\left(\rho \;\left(s\right)/\rho \;\left(s=\infty \right)\right)$ where ρ(s) is equilibrium (canonical) probability distribution of the solute to be at distance *s* from the surface. Adsorption free energy can be deduced from PMF by:(2)$$\Delta G_{ads}=-k_{B}T\left(\ln\frac{1}{\delta }\int_{r_{c}}^{r_{c}+\delta }ds\;e^{-W(s)/k_{B}T}\right)$$
where parameter δ is the thickness of the adsorption layer, $r_{c}$ is the closest approach of the molecule to the surface and $r_{c}+\delta$ is where the free region (unbound) starts, which can be determined from the distance at which W(s)=0. Note that while PMF depends on the specific choice of *s* (for example, distance to the center of mass of the solute or to a specific atom), the adsorption free energy does not depend on this choice.

We used classical molecular dynamics (MD) together with advanced sampling, metadynamics (MetaD) method [60] to calculate PMFs and the adsorption free energies of small biomolecules to ZnS nanomaterials. GROMACS 2021 [61,62] with PLUMED v.2.7 plugin [63,64,65] was used for the simulations. As a collective variable of MetaD we considered surface separation distance (SSD), which is defined as the minimum distance between the center of mass of a biomolecule to the nearest atom at the surface (both in slab and nanoparticle systems). A slightly modified definition of the minimum distance as implemented in the DISTANCES module with the LOWEST flag in PLUMED plug-in v 2.7 was used to provide a continuous derivative of the SSD over atomic coordinates. In MetaD simulation, the system is subjected to an external history-dependent bias potential which is a function of the collective variables (CV). The bias potential is represented as a sum of Gaussians applied along the CVs space to urge the system to explore configurations that have not yet been sampled. The potential of mean force can be calculated either from the accumulated bias potentials corrected by the distributions of CVs or by integration of the average force acting along the CV. It was shown previously that the force integration provides more stable and faster converging PMF compared to the standard or well-tempered metadynamics [56,66]. We have used the metadynamics method together with force integration to calculate the adsorption free energy of biomolecules to ZnS nanomaterials. The PMF is computed by:(3)$$W\left(s\right)=\int_{s_{n}}^{s_{0}}\langle F\left(s\right)\rangle ds+C$$
where *s* passes through all the relevant SSD values, 〈F(s)〉 is estimated by taking an average of the force applied along SSD excluding the contribution from the bias potential. The bias potential was updated every 1 ps by the addition of a Gaussian function of height 0.01 kJ/mol and width 0.05 Å to observe adsorption states in reasonable simulation time. The rate of deposition of Gaussians was chosen on the one hand to provide close to equilibrium (Hamiltonian) dynamics, on the other hand, to provide convergence of the algorithm in reasonable computational time.

Additionally, we introduced a fictional wall potential to prevent the adsorbent from visiting states in the bulk area that are far from the surface to reduce the time of convergence:(4)$$U_{wall}\left(s\right)=\kappa {\left(s-a\right)}^{4}$$

For all pristine systems, the wall was set at a=1.5 nm from the surface, and the energy constant, κ, was set to 40 kJ mol^−1^Å^−4^. For coated systems, the upper wall was set at a=3 nm to accommodate the coating layer.

In experimental studies, Langmuir isotherm is a standard model to estimate binding free energies. Since we have only one adsorbent binding to the surface, we can consider it as a diluted solution corresponding to the dilute limit of Langmuir isotherm. Experimental adsorption free energies are often defined with respect to the standard free energy, with the reference state corresponding to the concentration of solute 1M. Adsorption free energy computed according to Equation (Equation 2) corresponds to “excess” free energy, that is free energy change of transfer the solute from the bound region to the same volume far from the surface. By adding the term $k_{B}T\;\ln\left(c/c_{st}\right)$ to the calculated free energy one can re-estimate the free energy to any other defined standard state where *c* is the concentration of the molecule in the bound region of the simulation cell and $c_{st}$ is the concentration of the molecule in the standard state.

#### 2.3.2. Simulation Protocols

At the beginning of each simulation, the chosen adsorbate molecule was placed near the surface of the ZnS slab or nanoparticle, and the system was solvated with water molecules and 0.15 mM of Na and Cl ions. Each system was energy minimized for 4500 steps using the steepest gradient method and then pre-equilibrated for 200 ps in the NpT-ensemble (constant number of particle, pressure, and temperature) using Berendsen weak coupling [67] thermostat and barostat at 300 K and 1 bar. In the case of the slab system, semiisotropic NPT ensemble was used. Time constants for temperature and pressure coupling were 1 ps and 5 ps, respectively. The cut-off radius for short-range interaction (LJ) was 1.4 nm, and Particle Mesh Ewald (PME) method [68,69] was used in all simulations to treat the electrostatic interactions.

The metadynamics simulations were preequilibrated by a conventional MD system and performed initially for 2 ns under NVT conditions at 300 K to bring the molecule closer to the surface, after which the main part of the simulation started. The velocity-rescaling temperature coupling and relaxation time constant 1 ps were used. The Gaussian width (sigma) of 0.05 nm and height of 0.01 kJ·mol^−1^ have been applied for all simulations. The time needed for adsorption free energies to get converged was different for different biomolecules and in pristine and coated systems, varying from 100 ns to 2.5 μs for coated systems. In pristine systems, we excluded the first 50 ns from the force averaging, and ran each simulation for at least 300 ns. Statistical uncertainty was evaluated by block averaging taken each 50 ns. Additionally, the quality of the sampling was controlled by observing CV as a function of time and ensuring that the adsorbent sampled multiple times both bound states and states far from the surface. The time dependence of CVs for different cases is shown in Appendix A. In some cases, when the evolution of the CV showed poor sampling, the simulation time was prolonged to allow biomolecules to adopt different configurations on the surface of ZnS nanostructures and in particular to have time to find different binding sites on the ZnS nanoparticles. For coated systems, the first 200 ns of simulations were excluded from averaging, and the length of simulations was between 500 ns and 2.5 μs. The variance of the results found for every 50 ns of the whole trajectory was used to compute the statistical error.

## 3. Results and Discussion

### 3.1. Binding Preferences of Biomolecules to Pristine ZnS Nanosurfaces

We begin the analysis with results for the pristine ZnS surfaces. Figure 3 shows PMF curves for four molecules most strongly binding to pristine ZnS while PMFs of other molecules are shown in Appendix A. Numerical data on the adsorption free energies are gathered in Table 2. Both data on binding free energies and PMFs show that out of 29 biomolecules, 3 molecules (ASP, CYM, and GLU) exhibit significantly stronger adsorption to the ZnS (110) and nanoparticle. A weak d-glucose (DGL) binding can be observed at the plane surface, but no binding is seen for the spherical ZnS nanoparticle. All remaining adsorbents do not adsorb or have negligible adsorption energies on both nanostructures. Regarding the classification of biomolecules (hydrophobic, aromatic, polar, and charged), all strongly adsorbed biomolecules (ASP, CYM, GLU) are negatively charged, which is related to the polar behavior of ZnS nanostructures and accessibility of positively charged Zn atoms to bind with. Furthermore, in all systems of strongly adsorbed amino acids, the adsorption energy is higher in nanoparticles than in the homogeneous slab of ZnS (110). Our analysis (discussed in more detail in the section below) shows that this strong binding takes place at two-coordinated Zn atoms, which are present at a spherical surface but absent on plane surfaces. The carboxylic anionic group in ASP and GLU can better orient their oxygen atoms toward two-coordinated zinc atoms at the nanoparticle’s surface. This could be the reason for the significant difference in adsorption energy of ASP and GLU to ZnS slab (−2.9 and −2.5 kJ/mol) and nanoparticle (−30.3 and −39.1 kJ/mol). Cysteine anion also shows moderate adsorption to the ZnS plain surface (−3.7 kJ/mol) and strong adsorption to the ZnS nanoparticle (−31 kJ/mol).

The experimental adsorption of cysteine and serine on ZnS nanoparticles was studied by Hsu-Kim et al. [70]. It was found that cysteine, adsorbing on nanoparticles, induces negative zeta potential, which can indicate that it is the anionic form that binds to the surface. From data in Figure 5 of paper [70], experimental adsorption free energy of cysteine can be evaluated as −20 kJ/mol, which is similar to our result for nanoparticle surface. The absence of adsorption of serine to the ZnS surface observed in the same work is also consistent with our results.

### 3.2. Potential of Mean Force and Binding Configurations of Biomolecules at Pristine ZnS (110) Surface and Nanoparticle

We analyzed in more detail how the biomolecules bind to the planar slab and to spherical nanoparticle surfaces. The PMFs for the biomolecules with the strongest adsorption are shown in Figure 3, while PMFs for all biomolecules are presented in Appendix A. By investigating the positions and number of minima in these PMFs, we learn whether biomolecules bind directly to the surface, or through intermediate water molecules, or have multiple adsorption configurations on ZnS, as well as how the shape of ZnS nanostructures affects the binding modes of biomolecules.

First, we analyze the positions of minima in the PMFs to determine the binding configurations of biomolecules in both planar and spherical systems. As shown in Appendix A, in many cases PMFs have two minima, the first one in the range 0.3–0.4 nm and the other at 0.5–0.6 nm distance. Values of the PMF minima below 0.4 nm indicate that biomolecules directly bind to the surface, while minima at larger distances correspond to binding through the hydration shell and/or water-mediating mechanism. This is in accordance with Nawrocki et. al’s studies about the interaction of amino acids with ZnS (110) surface [42]. The global PMF minimum can correspond to either the first(direct) or second (water mediated) PMF minimum, depending on the type of amino acid and whether it binds to a planar or a spherical surface.

Considering strongly binding amino acids (Figure 3), we see that GLU and ASP have global PMF minima corresponding to direct binding in the case of the spherical surface, while in the case of planar surface strongest binding corresponds to a water-mediated mode. The situation is different for glucose, which shows the strongest binding to a planar surface in a direct mode of binding. Cysteine has the strongest direct binding mode both in the case of planar and spherical surfaces.

In order to understand large differences of strongly binding amino acids to spherical, respectively, planar surfaces, we analyzed in detail molecular configurations extracted from the MD trajectory considering cases when the sorbent was at the distance corresponding to the minimum of PMF. Typical configurations, including distances between important atoms, are shown in Figure 4. One can see that in the case of a planar surface, ASP and GLU bind to Zn atoms by only one of the oxygens of the COO${}^{-}$ group (Figure 4a,c), while they bind by both oxygen atoms to 2-coordinated Zn atoms, which are present at the edges of ZnS spherical nanoparticle (Figure 4e,f). Previously, Moloti et al. [71] suggested strong interaction of carboxylic oxygens of L-aspartic acid to ZnS nanoparticle from observation of the C-O peak appearance in the IR-spectra of L-aspartic acid-capped ZnS nanoparticle compared to pure L-aspartic acid. The difference in the adsorption free energy of ASP and GLU at slab and nanoparticle ZnS structures could be the result of difficulties to keep both oxygens at the planar surface while competing with water molecules at the surface. Of note, one can see ordered water structure in the first hydration layer at the planar ZnS surface (Figure 4a–d) while no such ordering is seen at the bent surface of the nanoparticle (Figure 4e–g). The difficulty in substituting strongly bound water from the first hydration layer is also likely the reason for the weaker binding of CYM to the planar surface compared to the nanoparticle where CYM can bind to 2-coordinated Zn atoms which do not have strongly bound water around.

Further analysis of PMFs in Figure 3 shows the PMF of ASP, CYM, and GLU have second local minima located at about 0.5 nm, indicating that these biomolecules have a weaker binding mode adsorbing through a mediating water molecule at the nanoparticle surface. The PMF of glucose (DGL) for the pristine ZnS slab has only one global minimum, showing that there is only one binding configuration on a ZnS planar surface.

Analysis of the PMF curves for weakly or non-adsorbing biomolecules (Appendix A) indicates two patterns: (1) systems with no minimum, and (2) systems with quite shallow minima in their PMF curve. The structures of these biomolecules are either strongly hydrophobic or have large hydrophobic groups together with electronegative atoms (O, N), which allow them to weakly adsorb to ZnS (at further distances around 0.5 nm) but their hydrophobic character and large size prevent them from penetrating the first water layer and adsorbing to ZnS directly. For instance, there are no minima in PMF of all hydrophobic biomolecules (ALA, ILE, LEU, MET, VAL), and most of the aromatic biomolecules, CYS, EST, ARG, and CHL. By looking into their structure, ALA, ILE, LEU, VAL, and PHE, have only H,C atoms in their structures and there is no adsorption to ZnS. MET and CYS could not reach the surface of ZnS and penetrate water layers since the sulfur atom in these molecules has low partial charge and electronegativity, which results in only weak electrostatic interactions with zinc atoms on the ZnS surface. ARG, PRO, TRP, and TYR have -OH or -NH groups but due to their size, these molecules need to displace more water to bind directly to the surface, which makes such binding unfavorable. In the next group, there are HID, HIE, ASN, EST, GAN, GLN, GLY, SER, THR, ETA, HIP, and LYS, which have a shallow minimum in their PMF. These biomolecules have -OH, -NH groups and they can make hydrogen bonds and are small enough to reach distances around 0.4 nm and have electrostatic interactions with ZnS. But these interactions are not strong enough compared to the interaction of water to the surface.

D-Glucose (DGL) is the only of the considered molecules which adsorb stronger to the ZnS slab and not the nanoparticle. For glucose molecules with more planar structure and several OH groups, it is difficult to find favorable orientation at the surface of the nanoparticle but it can attach to the ZnS (110) surface by orienting four of the -OH groups toward the surface S atoms and matching the surface structure (Figure 4). D-Glucose also has a weaker binding mode through the binding of OH-groups to the ordered water layer at the surface.

### 3.3. Binding Preferences of Biomolecules to PMMA-Coated ZnS (110) Slab and ZnS Nanoparticle

We investigated the adsorption of all considered in this paper biomolecules to a PMMA-coated ZnS slab, while for PMMA-coated nanoparticles, we investigated only several molecules that showed strong adsorption to ZnS pristine nanoparticles. The results presented in Table 2 show that the binding preferences of biomolecules to the coated ZnS slabs are very different from the pristine slabs. ASP, CYM, and GLU, which are adsorbed on pristine slab and even stronger to nanoparticles do not adsorb on the PMMA-coated slab nor to coated nanoparticles. But some other biomolecules (e.g., ILE, LEU, TYR), which do not adsorb on the pristine ZnS surfaces show rather a strong adsorption to the coated surface.

Figure 5 shows the adsorption free energy profiles of biomolecules having the strongest adsorption to PMMA-coated slab of ZnS (110) in comparison with the pristine slab, while for other molecules the adsorption profiles are shown in Appendix A. The PMFs have a wide, stretching up to 3 nm distance minimum, which approximately corresponds to the width of the PPMA coating layer. Insets in Figure 5 show radial distribution functions (RDF) between the center of mass of sorbent molecules and C-atoms of the PMMA backbone, which indicates strong affinity of these molecules to PMMA. The RDFs of non-adsorbing or weakly adsorbing molecules to PMMA presented in Appendix A show lower (or no) RDF maxima compared to the adsorbed biomolecules, with the general trend that the RDF of biomolecules with lower adsorptions to the coated slab have more diffuse and lower RDF peaks. This suggests that the binding properties of molecules to a coated surface are defined by the affinity of these molecules to the coating and not to the nanosurface/nanoparticle itself, though further investigation on the mutual role of the surface and coating may be needed to come to a definite conclusion.

The molecules that adsorb strongly on PMMA-coated surfaces are typically aromatic or hydrophobic molecules such as ILE, LEU, MET, PHE, TRP, TYR, and VAL. Computations of binding energies of several peptides to PMMA resin were performed in recent work by Iwasaki et al. [72], where it was found that some amino acids including TRP and PRO show strong binding affinities to the methoxycarbonyl groups and methyl groups of PMMA.

Figure 6 compares PMF of ASP, CYM, and GLU in pristine form and coated form of ZnS nanoparticle. The adsorption strength of these biomolecules decreased strongly by coating the nanoparticle with PMMA. The RDF profiles of biomolecules to 3-MMA monomer have no peak and show the least affinity of these anionic biomolecules to the polymer (Appendix A). The number density profile of PMMA in Figure 7 shows that there is a sharp peak at around 0.19 nm, which corresponds to oxygen atoms of PMMA binding to zinc atoms and occupying these binding sites hindering oxygen atoms in ASP and GLU and sulfur atom of CYM to bind directly to the surface. The PMFs show a minimum around adsorbing distance even for coated surface, but they have a positive value due to the need for adsorbing molecule to displace oxygen atoms of PMMA, and this causes the adsorption energy to decrease dramatically.

### 3.4. Influence of Solvation Shell on Binding Preferences of Biomolecules to ZnS (110) Slab and ZnS Nanoparticle

To obtain additional insight into molecular mechanisms governing the adsorption of biomolecules on ZnS surfaces and their preferences, we investigated the structural properties of water molecules surrounding the ZnS (110) slab and ZnS nanoparticle both in pristine and PMMA-coated forms. Figure 7a shows density profile of water molecules, Na${}^{+}$ and Cl${}^{-}$ ions, and PMMA around ZnS slab. The slab is coated by PMMA molecules only from one side while the other side of the slab remains pristine. For this reason, the left side of Figure 7a shows the density profile of water and ions at the pristine ZnS surface, while the right side shows the density profiles of water, ions, and PMMA at the coated surface.

The water density profile at the pristine ZnS slab surface (left part of Figure 7a) shows several sharp maxima illustrating highly ordered water layers. Such ordered water layers were observed in simulations of water near other highly polar surfaces such as TiO${}_{2}$ [73] and ZnO [74]. In the case of the PMMA-coated surface (right part of Figure 7a), only the first peak of water remains high, while other peaks are strongly suppressed due to the presence of coating. In the case of the coated planar surface, PMMA makes about a 2 nm thick layer almost completely expelling water from it.

For the pristine nanoparticle surface (Figure 7b) the number density profile of water molecules shows some structure in the range of 2.5–3.5 nm from the nanoparticle center (or within 1 nm from the surface) and then approaching the bulk water density. This indicates that the pristine ZnS nanoparticle has a solvation shell that is about 1 nm thick. For the coated nanoparticles, the water is depleted from the coating region, but to a less extent than in the case of the plain surface. Also, the PMMA density decreases faster with the distance compared to the plane surface case. This can be expected as available volume is increasing with the distance as $r^{2}$ for a spherical nanoparticle but remains constant for a plain surface. Density profiles relative to the nanoparticle’s center may be somewhat diffused since nanoparticle surface atoms are located at slightly different distances from the nanoparticle center. For a more clear presentation of the hydration layers, we also show in Figure 7c the density of water oxygen atoms as a function of distance to the nearest atom of the pristine nanoparticle. This density profile has well-expressed peaks corresponding to the water structuring in several hydration layers near the surface.

The distribution of water molecule orientations in the first solvation shell (taken up to the first minimum of the corresponding density profile) of the ZnS plain and spherical nanosurfaces is shown in Figure 7d–h. Since the biomolecules need to penetrate the water layers, the orientation of water molecules close to nanoparticles is very important for the biomolecules binding [75]. Here, we measured the orientation in terms of two angles: θ is the angle between the nanosurface normal or nanoparticle radial direction and the dipole moment of the water molecules, while α is the angle between the nanosurface normal or nanoparticle radial direction and the OH bond of the water molecules, as illustrated in Figure 7f. Figure 7d,e show distributions of these angles in the solvation shell of the planar surface. The distribution of the dipole vectors has two maxima at about 65${}^{\circ }$ and 115${}^{\circ }$. These characteristic orientations of water are seen in the snapshots in Figure 4, where water molecules in the first layer and coordinated to Zn atoms have a typical orientation of dipole vector of 65${}^{\circ }$ while water molecules coordinated to S atoms have dipole angle about 115${}^{\circ }$. The angular distribution of water around spherical nanoparticles is less structured. The maxima of these distributions indicate that the water molecules in the nearest solvation shell surrounding the ZnS nanoparticles are more often oriented to the surface by their hydrogen atoms.

The structural details of the hydration layers give further insight into biomolecular adsorption to ZnS surfaces. Small biomolecules (or binding groups of larger biomolecules) bind typically at 0.2–0.3 nm distance from the ZnS surface (Appendix A), the same distances where the first tightly bound solvation shell of water is located (Figure 7). For the amino acids to attach directly to the nanoparticle, the water molecules in the solvation shell must be replaced. Furthermore, for some biomolecules binding to the surface may change the orientation of surrounding water molecules. The stability of the bound state is influenced by changes in the density and the preferred orientation of water molecules near the solutes and varying them affects the system’s free energy [76,77,78]. Highly polar surfaces such as ZnS form a strongly bound water layer which is characterized by high peaks in the positional and orientational distributions of water molecules. Displacement of water molecules from the hydration layer gives an unfavorable contribution to the binding free energy, which explains low (or positive) binding free energy for the most of considered molecules.

All biomolecules showing strong adsorption to ZnS are negatively charged. These anionic biomolecules adsorb through their anionic carboxyl or sulfur groups to Zinc atoms at the surface of nanoparticles where they are able to compete with water molecules due to favorable electrostatic interactions. Furthermore, in the case of spherical nanoparticles the biomolecules prefer to adsorb to the zinc atoms at the edges where the water molecules are not ordered enough and where it is easier for an anionic molecule to reach zinc atoms which are two coordinated (Figure 4). The positively charged LYS and HIP, while having favorable electrostatic interactions with negatively charged sulphur atoms, do not adsorb to them likely because of larger cationic groups, as well as due to the strong binding of H-atoms of water to S atoms of the surfaces.

## 4. Conclusions

In this work, we investigated the effect of surface curvature and coating on the adsorption behavior of 29 small adsorbents, which are building blocks of different biomolecules to the pristine and PMMA-coated ZnS (110) planar surfaces and to ZnS spherical nanoparticles using molecular dynamics simulations and Metadynamics method. We found that only a few negatively charged molecules (ASP, CYM, GLU amino acid side chains) show relatively strong binding to ZnS planar surfaces, although some polar and planar adsorbents such as glucose also showed a weak adsorption tendency. We have further shown that the shape of the ZnS nanomaterials, that is planarity vs curvature of the surface, affects the adsorption energy significantly. Small-sized nanoparticles were found to provide stronger adsorption of anionic molecules, primarily by their binding to lower-coordinated Zn atoms at the edges between different facets of the spherical nanoparticle, while d-glucose was found to adsorb to a planar surface but not to a spherical shape. Our studies of the coating effect on the adsorption behaviors revealed that the presence of coating and its chemical properties affect drastically the binding of biomolecules. Thus, many of the considered molecules were found to bind stronger to ZnS nanosurfaces by binding to the coating, while the biomolecules, which show strong binding to a pristine surface were found not adsorbing to the coated slabs and nanoparticles.

The analysis of binding preferences of 29 small essential biomolecules to ZnS (110) slab and ZnS nanoparticle suggests a way to design peptides that bind to ZnS nanostructures and control nanostructure shapes or recognize nanoparticles with certain shapes. We found three amino acids (ASP, CYM, GLU), which exhibit strong binding to the curved (spherical) shape of ZnS. These amino acids could be used to prepare peptides that bind to ZnS with specific shapes stronger than others. In fact, cysteine-rich peptides are used to fabricate peptides-coated ZnS surfaces for various applications [79]. Furthermore, the modification of the surface of ZnS nanomaterials can change the adsorption affinity and selectivity of different biomolecules, which can be favorably used in various biomedical applications.

## Figures and Tables

**Figure 1 nanomaterials-13-02239-f001:**
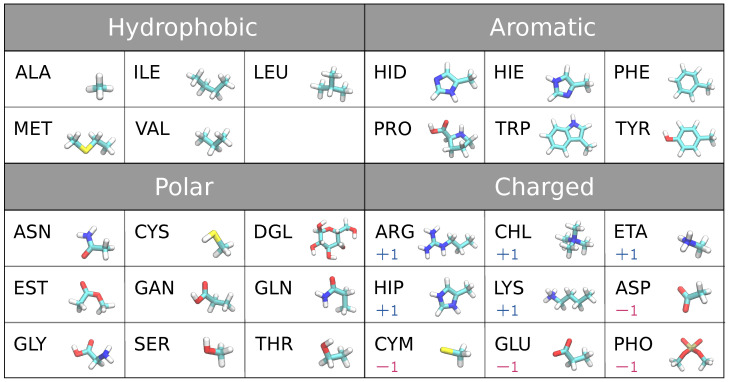
29 small molecules (adsorbates) classified as hydrophobic, aromatic, polar, charged ±1 molecules.

**Figure 2 nanomaterials-13-02239-f002:**
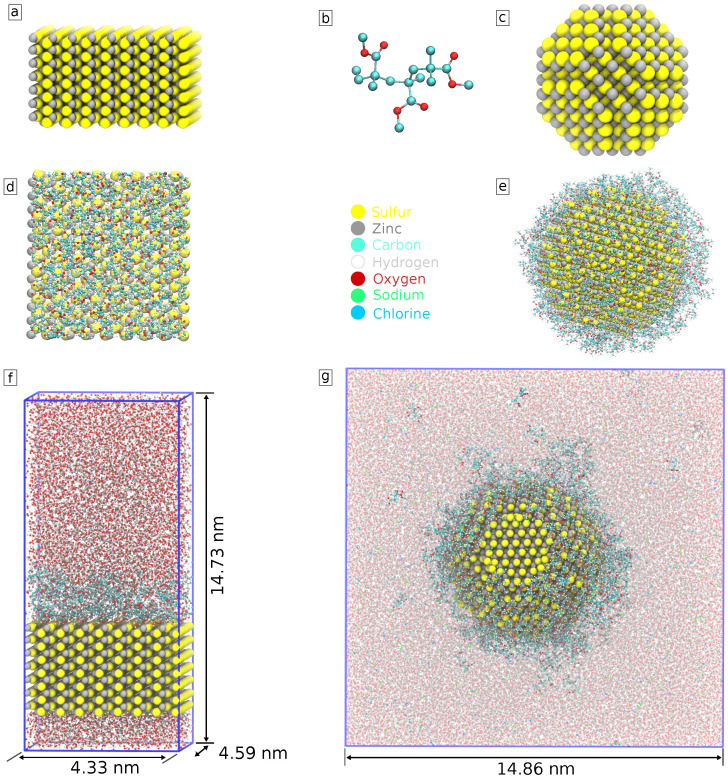
Model of (**a**) ZnS (110) slab; (**b**) PMMA; (**c**) ZnS nanoparticle; (**d**) coated slab of ZnS (110) (top view); (**e**) coated ZnS nanoparticle. The system used to simulate (**f**) coated with PMMA ZnS (110) slab and (**g**) coated with PMMA nanoparticle in the solution of water and ions.

**Figure 3 nanomaterials-13-02239-f003:**
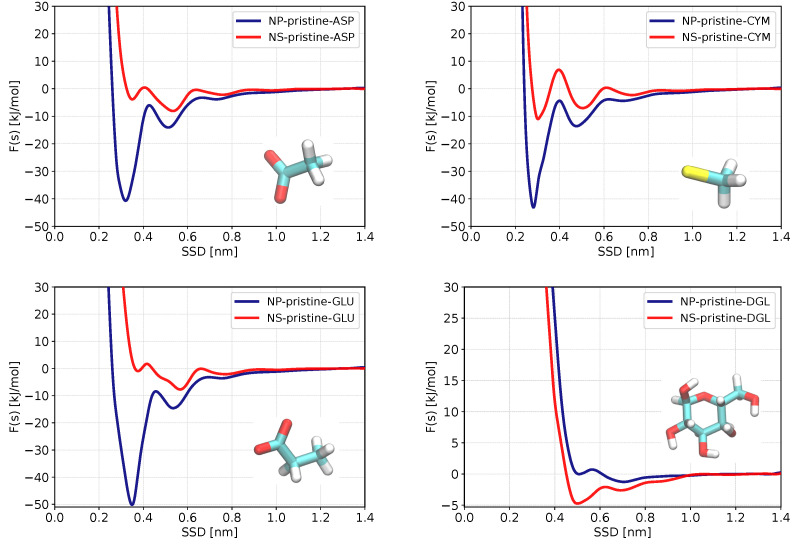
Adsorption free energy profiles as functions of the surface separation distance (SSD) for biomolecules with the strongest adsorption on ZnS (110) slab (red) and ZnS nanoparticle (blue) in pristine (non-coated) form.

**Figure 4 nanomaterials-13-02239-f004:**
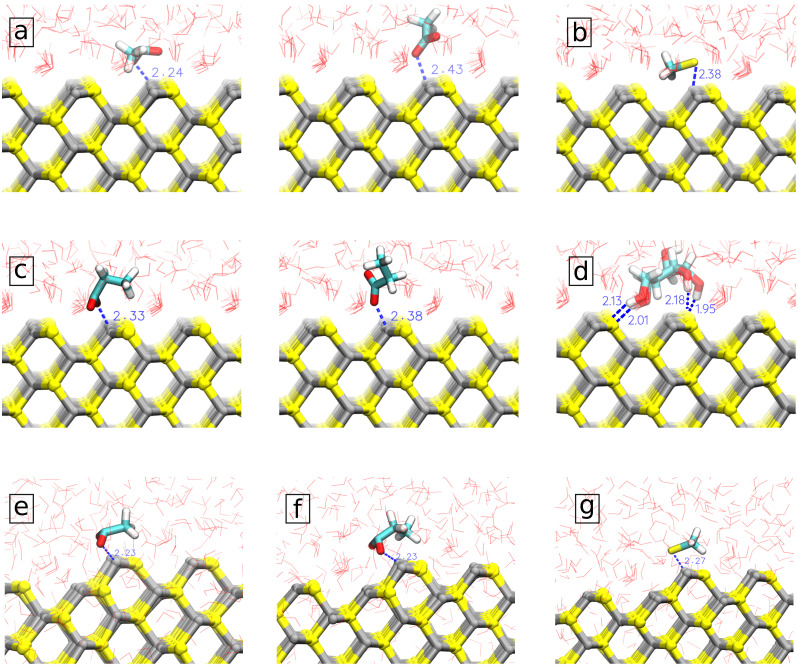
Molecular orientation and surface separation distance (SSD) of strongly adsorbed biomolecules (**a**) ASP; (**b**) CYM; (**c**) GLU; (**d**) DGL) on the pristine ZnS (110) slab and strongly adsorbed biomolecules; (**e**) ASP; (**f**) CYM; (**g**) GLU on the pristine ZnS nanoparticle.

**Figure 5 nanomaterials-13-02239-f005:**
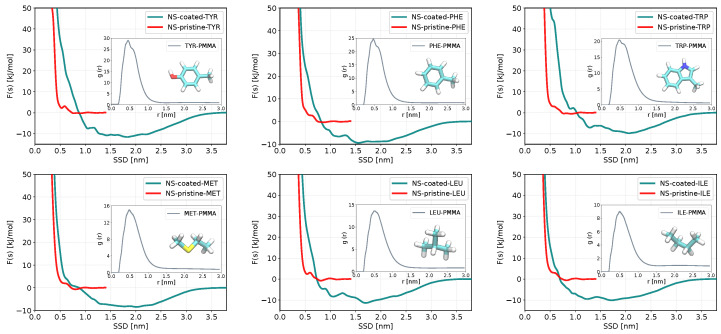
The outer plots represent adsorption free energy profiles as functions of the surface separation distance (SSD) for biomolecules with the strongest adsorption on the coated slab of ZnS (110) (green) and their corresponding profiles in pristine form (red). The RDF of each biomolecule to a PMMA molecule is shown in the inner plots.

**Figure 6 nanomaterials-13-02239-f006:**
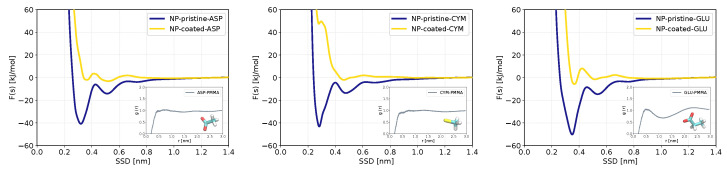
Free energy profiles as functions of the minimum distance between the center of mass of each biomolecule to the nearest C atom of the PMMA molecule (outer plots). The RDFs of each biomolecule to PMMA are shown in the inner plots.

**Figure 7 nanomaterials-13-02239-f007:**
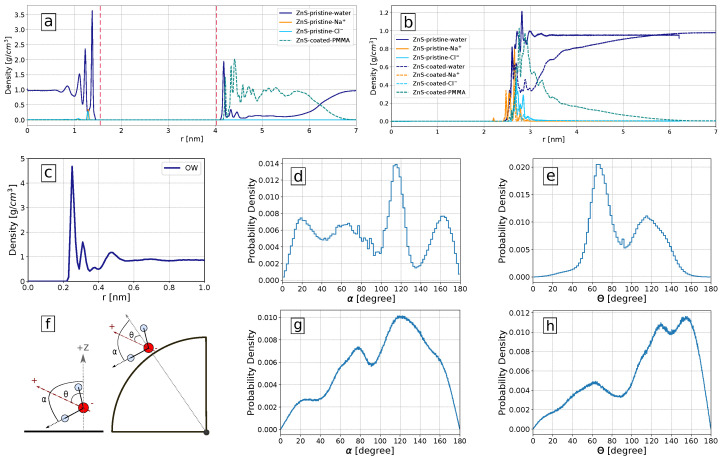
Solvation properties of ZnS surfaces: (**a**) The density profile of water, Na${}^{+}$, Cl${}^{-}$, and PMMA around pristine (left part) and coated ZnS (110) slab surface (right part), the position of the outmost Zn atoms are shown as vertical dashed lines; (**b**) density profile of water, Na${}^{+}$, Cl${}^{-}$ and PMMA relative to the center of mass of pristine (solid lines) and coated (dashed lines) ZnS nanoparticles; (**c**) Density of oxygen atoms of water molecules around the pristine nanoparticle with respect to the distance to the nearest surface atom of the nanoparticle; (**d**,**e**) Probability density of α and θ angles with respect to the z-axis (perpendicular to the surface of the slab) in the pristine slab; (**f**) visual representation of θ and α angles representing the orientation of the water dipole moment and the water OH bonds, respectively, in a slab (left) and nanoparticle (right); (**g**,**h**) Probability density of α and θ with respect to the radial direction of the pristine nanoparticle.

**Table 1 nanomaterials-13-02239-t001:** ZnS and PMMA interaction parameters.

**Bond potential**	**k_b_ (kJ/mol)**	**r_0_ (Å)**	
Zn–S	92,000	1.6	
**Angular potential**	$k_{\theta }$ **(kJ/mol)**	$\theta_{0}$ **(** ${}^{\circ }$ **)**	
S–Zn–S	274.022	109.47	
Zn–S–Zn	274.022	109.47	
**Non-bonded**	**Charge (e)**	***σ* (Å)**	***ε* (kJ/mol)**
Zn	2.0	3.816	0.022
S	−2.0	4.27	1.087
**Special LJ parameters**			
Zn *-O${}_{C}$		1.75	86.00
S *-O${}_{C}$		5.80	0.01

* Surface ZnS atoms interacting with oxygen atoms of carboxyl PMMA group.

**Table 2 nanomaterials-13-02239-t002:** The adsorption free energy of biomolecules on pristine ZnS (110), coated ZnS (110), pristine ZnS nanoparticle, coated ZnS nanoparticle. Statistical error is not shown if it is below 0.1 kJ/mol.

Adsorbate	Code	NS-Pristine	NS-Coated	NP-Pristine	NP-Coated
SCA of alanine	ALA	0.2	−2.1 ± 0.6	0.7	–
SCA of arginine	ARG	0.1 ± 0.1	−0.3 ± 0.2	0.8	–
SCA of aspargine	ASN	0.0 ± 0.1	−0.2 ± 0.4	0.7	–
SCA of aspartic acid	ASP	−2.9 ± 0.2	0.3 ± 0.2	−30.3 ± 2.3	−0.4 ± 0.4
SCA of cysteine ion	CYM	−3.7 ± 0.7	0.4 ± 0.1	−31.1 ± 7.4	0.2
SCA of cysteine	CYS	0.0	−3.2 ± 0.6	0.6	–
SCA of glutamine	GLN	0.1	−1.0 ± 0.4	0.7 ± 0.1	–
SCA of glutamic acid	GLU	−2.5 ± 0.1	−0.5 ± 0.2	−39.1 ± 4.8	−1.0 ± 4.6
SCA of histidine	HID	0.0 ± 0.1	−2.8 ± 0.3	0.6	–
SCA of histidine	HIE	0.0 ± 0.1	−3.0 ± 0.5	0.5	–
SCA of histidine	HIP	0.1 ± 0.1	0.2 ± 0.1	0.8	–
SCA of isoleucine	ILE	0.0	−7.4 ± 0.9	0.7	–
SCA of leucine	LEU	0.2	−7.7 ± 0.1	0.6	–
SCA of lysine	LYS	0.3	0.4 ± 0.1	0.8	–
SCA of methionine	MET	0.1	−5.9 ± 0.7	0.6	–
SCA of phenylalanine	PHE	0.2	−6.6 ± 0.7	0.6	–
SCA of serine	SER	0.3	−0.4 ± 0.5	0.7	–
SCA of threonine	THR	0.2	−1.3 ± 0.4	0.7	–
SCA of tryptophan	TRP	0.0	−6.6 ± 0.8	0.5 ± 0.1	–
SCA of tyrosine	TYR	0.3 ± 0.1	−8.7 ± 1.1	0.6	–
SCA of valine	VAL	0.0	−4.7 ± 0.8	0.6	–
SCA glutamic acid (neutral)	GAN	0.0 ± 0.1	−1.9 ± 0.4	0.5	–
SCA of glycine (amino acid)	GLY	−0.8 ± 0.1	−0.3 ± 0.1	0.2	–
proline (amino acid)	PRO	0.1 ± 0.1	−2.5 ± 0.4	0.3 ± 0.1	–
choline group of lipid	CHL	0.0 ± 0.1	0.4 ± 0.1	0.9	–
phosphate group of lipid	PHO	−1.1 ± 0.2	0.0 ± 0.2	−1.5 ± 0.2	−0.4
etanolamine group of lipid	ETA	0.2 ± 0.3	0.4 ± 0.1	1.3 ± 0.2	–
ester group of lipid	EST	0.0	−2.6 ± 0.4	0.5 ± 0.1	–
D-glucose	DGL	−1.7 ± 0.2	−2.0 ± 0.5	0.2 ± 0.1	–

## Data Availability

Data are available at the institutional repository.

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
