# Peer review of "Biomolecular Adsorprion at ZnS Nanomaterials: A Molecular Dynamics Simulation Study of the Adsorption Preferences, Effects of the Surface Curvature and Coating"

_nanomaterials, 2023, doi:10.3390/nano13152239_

Round 1

Reviewer 1 Report

The well-written research by Rahmani and Lyubartsev focused on the adsorption of different small molecules on pristine and PMMA-coated surfaces of zinc sulfide (ZnS). The motivation, problem, and purpose of the study are clearly discussed in the Introduction. The obtained results are novel and interesting. However, the paper might be accepted in Polymers after minor issues could be answered.

1. From line 206, it is unclear which partial charge calculation method was used (RESP or AM1-BCC) for the partial charge calculation. Additional information should be added to the manuscript.

2. In the caption of Figure 6, the last sentence refers to the inner plots, although they are not depicted in the figure.

Author Response

We thank reviewer for the careful reading the paper and giving comments. We revised the manuscript accordingly. Below we give point-to-point answers to the reviewer comments:

Reviewer:

"1. From line 206, it is unclear which partial charge calculation method was used (RESP or AM1-BCC) for the partial charge calculation. Additional information should be added to the manuscript."

Author answer: 

We have used AM1-BCC method for partial change calculations which is default in the acpype utility. We comment this in the revised text

Reviewer:

"2. In the caption of Figure 6, the last sentence refers to the inner plots, although they are not depicted in the figure."

We apologize for his miss. In the revised manuscript we update the Figure with inner RDF plots

Reviewer 2 Report

This paper systematically investigated the adsorption of 29 types of common biomolecules on the surface of ZnS nanomaterials with the effects of surface curvature and coating being carefully explored. The simulation setup is reasonable and the simulation results are abundant. By calculating and comparing the potential of mean forces of the surface adsorption for different types of biomolecules, the molecules that have the strongest binding strength to the ZnS surface have been identified. In addition, the underlying mechanisms provided for the understanding of the adsorption behaviors of these biomolecules are sound and inspiring. Overall, this work is well-written and helpful in deepening the understanding of the adsorption of biomolecules on polarized surfaces. Hence, it is recommended this work can be published in the journal of Nanomaterials.

Author Response

We thank reviewer for careful reading of our manuscript and positive evaluation

Reviewer 3 Report

The aim of this work is to provide a detailed description of biomolecular interaction with pristine and PMMA-coated ZnS nanosurfaces using enhanced-sampling molecular dynamics. This work is praiseworthy for its rigorous experimental data and wealthy theoretical explanations. The author should add the following information to the manuscript before publication.

1.      How does the author distinguish between physical adsorption and chemical adsorption?

2.      What are the shortcomings of molecular dynamics simulation in studying adsorption?

3.      How does the author determine that the system has reached a saturated adsorption state?

4.      Two relative papers are suggested to be cited (10.1021/acsami.1c22035, 10.1016/j.xphs.2022.05.012).

5.      Finally, the grammar of English needs further modification.

Moderate editing of English language required.

Author Response

We thank reviewer for careful reading of the manuscript and comments made.
Below our answers to the referee comments and description of changes made
in the manuscript:

Reviewer:
1.      How does the author distinguish between physical adsorption and
chemical adsorption?

Author answer:
In this paper we adress to the questions of physical adsorption of
biomolecules to the ZnS surfaces. We use classical molecular dynamics where
bonds between the atoms are predetermined and cannot change during the
simulations. Adsorption of biomolecules to nanopartocles goes in most
cases through physisorption mechanisms.   Physical adsorption takes place
due to electrostatic and/or van-der-Waals interactions which are well
described by classical molecular dynamics. We comment on this in page 3
(lines 102-105) of the revised manuscript.

Reviewer:
2.      What are the shortcomings of molecular dynamics simulation in
studying adsorption?

Authors answer:

The known shortcoming of molecular dynamics simulations is that results
can depend on the choice of force field, which is always an an approximaton,
and in some cases may be not properly parameterized. Furthermore, in some
cases computations may require to long time to converge.We discuss the choice
of forcefield and its parameterization in Section 2.2 of the manuscript,
and we describe methodology to ensure convergence and evaluate statistical
error in page 8, and providing data on evolution of collective variables in
Supporting Information. We also refer to the paper by Machaelis et at (new ref 41) where challenges and limitations of molecular dynamics to study biomolecular adsorption are discussed, adding a note in lines 113-114

Reviewer:
3.      How does the author determine that the system has reached a saturated
adsorption state?

Authors answer:

In this paper  we analyse low-concentration limit of adsorption described
by the linear part of Langmuir model, and do not address to a saturated state.
We explain this in page 8  (from line 266 ) of the revised manuscript.

Reviewer:
4.      Two relative papers are suggested to be cited (10.1021/acsami.1c22035, 10.1016/j.xphs.2022.05.012).

Author answer:

We cite these paper in the revised manuscript, new refs 39,40 

Reviewer:

5.      Finally, the grammar of English needs further modification.

Author answer:

We have corrected grammar in sentences where we found errors

Reviewer 4 Report

The authors study the adsorption of a set of biomolecules on ZnS surfaces from aqueous saline solution using molecular dynamics simulations and free energy calculations. Both flat surface slabs and a nanoparticle of ZnS are considered. These simulations are repeated for PMMA coated ZnS surfaces.

This manuscript is well written and gives good insights on the adsorption of these standard biomolecules onto the ZnS surface. This material and the PMMA coated version are good candidates for use in quantum dots and so this study is highly relevant.

After a number of comments listed below are addressed, I can recommend this manuscript for publication.

- Have simulations been conducted on PMMA on another surface? Given the thickness of the PMMA surface, does the underlying ZnS surface have any physiochemical effect on the adsorption?

-The partial charges of -2 and +2 chosen for S and Zn, respectively, in the ZnS phase may be too large and lead to an overestimation of Coulombic interaction strengths. This is particularly relevant since the largest free energy of adsorption on the surfaces was for the negatively charged ASP, CYM, and GLU species. A polarizable model for electrostatic interactions (as used in the original force field) would likely partly compensate for the large magnitude of charge and Zn and S, but a fixed charge model would likely overestimate electrostatic interactions.

-In Eq. (1) the free energies are written as G, but in the next sentence, they are written as F. Please use a consistent notation, unless the difference between Gibbs and Helmholtz free energies is implied, in which case, this should be mentioned. 

-The RDFs mentioned in the caption are not show in Figure 6 in the PDF version of the file I opened.

Author Response

We thank reviewer for careful reading of the manuscript and comments made.
Below our answers to the referee comments and description of changes made
in the manuscript:

Reviewer:
Have simulations been conducted on PMMA on another surface? Given the
thickness of the PMMA surface, does the underlying ZnS surface have any
physiochemical effect on the adsorption?

Author answers:
No, we have done simulations with PMMA only on ZnS surface. We discuss in
the text (section 3.3), refering to the obtained data, that adsorption to the
coated nanosurface / nanoparticle is determined mostly by the adssorbent - PMMA interactions. This is suppoted by the correlation between strength of
adsorption and adsorbent-PMMA RDFs. It is likely that underlying surface have
minor effect, but quantitative  investigation of the role of the underlying
surface (with the same coating) may be a matter of further  investigation, and
we added note on this in lines 431-433 of the revised manuscript.

Reviewer:
-The partial charges of -2 and +2 chosen for S and Zn, respectively, in the
ZnS phase may be too large and lead to an overestimation of Coulombic
interaction strengths. This is particularly relevant since the largest free
energy of adsorption on the surfaces was for the negatively charged ASP,
CYM, and GLU species. A polarizable model for electrostatic interactions
(as used in the original force field) would likely partly compensate for
the large magnitude of charge and Zn and S, but a fixed charge model would
likely overestimate electrostatic interactions.

Authors responce:
We used charges -2 and +2 as it was the case of the original force field.
Since in our model the positions of Zn and S atoms are mostly fixed by the
bonded interactions on a  regular structure, the electrostatic field of the
atoms is largerly cancel already on a few Ångström from the surface.
Still, in the original force field polarization term (Core-shell model, or Drude oscillator) was used only on S atoms of ZnS, and it was introduced with the main purpose to provide correct ZnS structure within non-bound force field, while
surrounding molecules were described by the classical fixed charged force
field. We therefore believe that polarization would not have major effect
on the adsorption results, and we chose non-polarizable model due to its
much higher efficiency in computations.

Reviewer:
-In Eq. (1) the free energies are written as G, but in the next sentence,
they are written as F. Please use a consistent notation, unless the
difference between Gibbs and Helmholtz free energies is implied, in which
case, this should be mentioned. 

Authors:
We are  sorry, this was missprint. We denote (Gibbs) free energy as G.

Reviewer:
-The RDFs mentioned in the caption are not show in Figure 6 in the PDF
version of the file I opened.

Author reply:
We apologize for this. In the revised manuscript we updated Figure 6
with inner plots showing RDFs.

Round 2

Reviewer 3 Report

Accept in present form